# Deep Neural Network for the Detections of Fall and Physical Activities Using Foot Pressures and Inertial Sensing

**DOI:** 10.3390/s23010495

**Published:** 2023-01-02

**Authors:** Hsiao-Lung Chan, Yuan Ouyang, Rou-Shayn Chen, Yen-Hung Lai, Cheng-Chung Kuo, Guo-Sheng Liao, Wen-Yen Hsu, Ya-Ju Chang

**Affiliations:** 1Department of Electrical Engineering, Chang Gung University, Taoyuan 333, Taiwan; 2Department of Biomedical Engineering, Chang Gung University, Taoyuan 333, Taiwan; 3Neuroscience Research Center, Chang Gung Memorial Hospital, Linkou, Taoyuan 333, Taiwan; 4Department of Neurology, Chang Gung Memorial Hospital, Linkou, Taoyuan 333, Taiwan; 5School of Physical Therapy and Graduate Institute of Rehabilitation Science, College of Medicine, and Health Aging Research Center, Chang Gung University, Taoyuan 333, Taiwan

**Keywords:** fall, physical activity, foot pressures, inertial sensing, deep learning, deep neural network

## Abstract

Fall detection and physical activity (PA) classification are important health maintenance issues for the elderly and people with mobility dysfunctions. The literature review showed that most studies concerning fall detection and PA classification addressed these issues individually, and many were based on inertial sensing from the trunk and upper extremities. While shoes are common footwear in daily off-bed activities, most of the aforementioned studies did not focus much on shoe-based measurements. In this paper, we propose a novel footwear approach to detect falls and classify various types of PAs based on a convolutional neural network and recurrent neural network hybrid. The footwear-based detections using deep-learning technology were demonstrated to be efficient based on the data collected from 32 participants, each performing simulated falls and various types of PAs: fall detection with inertial measures had a higher F1-score than detection using foot pressures; the detections of dynamic PAs (jump, jog, walks) had higher F1-scores while using inertial measures, whereas the detections of static PAs (sit, stand) had higher F1-scores while using foot pressures; the combination of foot pressures and inertial measures was most efficient in detecting fall, static, and dynamic PAs.

## 1. Introduction

Increasingly, research and discussion have focused on the recognition of physical activity (PA) and fall detection in recent years [1,2]. PA interventions have been evidenced to reduce the risks of cardiometabolic syndrome, falls, depression, anxiety, and dementia [3]. PA interventions have also attracted increasing interest for their potential health benefits in various diagnostic populations [4]. Several studies have even developed disease-related PA markers to differentiate the distribution of activity levels in individuals with chronic cardiovascular disease from healthy individuals [5]. On the other hand, fall detection provides a practical solution for fall scenarios, calling for help automatically if they occur. In particular, falls are a debilitating problem among the elderly [6] and individuals with Parkinson’s disease [7], multiple sclerosis [8], and stroke [9]. Usually, a fear of falling restricts an individual’s participation in daily activities [10,11,12]. The incorporation of fall detectors into mobile assistive technology may enable older people to live independently at home [13,14].

Currently, vision-based [15,16], inertial sensing-based detections [1,2], and hybrid frameworks [17,18] are the major approaches for PA monitoring and fall detection. The vision-based approaches use single or multiple cameras to capture human postures or movements [19]. Rapid changes in human posture are linked with fall incidence [20,21], whereas specific postures or patterns of human movements are present in various types of PAs [15]. Recently, several advanced methods based on deep neural networks have been proposed to extract the discriminant patterns related to human postures or movements from images through machine learning [22,23,24]. Vision-based approaches are beneficial due to the characteristic of wear-free, unobstructed monitoring, but it is only available in spaces where cameras have been installed.

In contrast to the restrictions of the vision-based approaches, the inertial-based approaches attach wearable devices to the human body or limbs to record the static and dynamic activity of daily life (ADL) anywhere. Popular wearable devices, such as Fitbit^®^ and JawboneUP^®^, employ an accelerometer to estimate activity levels, sleep quality, and energy expenditure. Recently, wearable PA monitors have undergone extensive development by attaching different varieties of sensors to the body to capture human motion data, from which several detection algorithms for various kinds of PAs have been developed. In these, inertial sensing based on accelerometers [25,26,27], paralleled with gyroscopic measurements [28,29], or further combined with magnetic field measurements [30,31], are used to detect various PAs.

Using similar sensor technology, accelerometers or gyroscopes are mostly used to detect falls during PAs. A great acceleration caused by the impact on the ground provides a direct marker for detecting falls [32,33,34,35,36,37]. Posture identification after impact can be used to reduce the false positive rate of fall detection [38,39,40,41]. In addition, a fall event can be characterized by the increasing angular velocity that peaks at the moment a knee or hip hits the ground [42]; therefore, gyroscopic measurements provide additional criteria for fall detection [43,44].

In recent years, deep learning neural networks have provided new ways to detect falls during PAs or classify various types of PAs. A convolutional neural network (CNN) is used to capture the discernible features from the inertial data. The succeeding fully connected network is then used as a classifier. The CNN and fully connected network weights are derived from machine learning based on the labeled data. The CNN-based model has been used to detect falls during PAs [45,46,47] and classify various types of PAs [48,49,50,51,52].

On the other hand, recurrent neuronal networks (RNNs) can aggregate the features of the inertial data at various time points to capture the sequential pattern of the fall or physical activity. The aggregated features are then used to detect the falls during PAs [48,53,54] and classify various types of PAs [55] through a fully connected network. In addition, a hybrid approach uses CNN in the first layer to extract the temporal features from the raw inertial data and applies RNN in the second layer to detect falls during PAs [56,57,58] and classify various types of PAs [59,60,61].

State-of-the-art PA classification and fall detection are illustrated in Table 1 and Table 2, respectively. Most of these studies placed the sensors on different parts of the body, including the waist, chest, back trunk, wrist, arm, leg, etc. Each placement has unique advantages in practical applications, e.g., sensors can be easily attached to certain areas, or it is easier to capture the main body motion or distinct features of specific movements using other areas. On the other hand, placing such sensors in shoes is also beneficial for detecting lower-extremity movements. A study attached an inertial measurement unit (IMU) containing an accelerometer and a gyroscope in a subject’s left shoe to detect different walking conditions (level walking, descending and ascending slope walking, walking downstairs, and walking upstairs) using a prediction model based on dorsi–plantar flexion angular velocities or a CNN based on raw inertial data [62]. Another study used a combination of inertial sensing on the trunk and thigh and foot pressure to classify postural transitions, locomotion, and walking type using a decision tree [63]. As for fall detection, one study used the embedded accelerometer in shoes to detect falls during PAs using empirical rules based on acceleration magnitude and the estimated posture [64]; another study employed an algorithm based on waist acceleration and the center of plantar pressure to detect falls during PAs [65].

As shoes are commonly worn during off-bed activities, they are worn during most outdoor activities. However, this methodology is rarely adopted for PAs and fall detection in the literature. In this work, we focus on the detection of falls and various types of PAs in parallel via footwear sensing and deep-learning technology. A novel footwear sensing system has been developed for instrumented shoes that are equipped with 11 force-sensitive resistors (FSRs) in the insole to measure foot pressures, and an IMU with an accelerometer, a gyroscope, and a magnetometer to measure foot inertial dynamics on each side. A novel deep-learning neural network based on a hybrid CNN and RNN model was used to learn the discernible features from the inertial signals and foot pressures and detect simulated falls and various types of PAs. To validate the feasibility of the proposed footwear approach, we define four kinds of simulated falls and seven types of activities of daily living, including jumping, jogging, level walking, walking downstairs, walking upstairs, sitting, and standing. The detections based on various combinations of signals and parameters were validated and compared based on a dataset collected in this study.

## 2. Materials and Methods

### 2.1. Subjects and Experimental Activities

A sample of 32 healthy participants (16 males and 16 females; age of 21.5 ± 2.0 yrs; height of 167.2 ± 7.1 cm; weight of 62.4 ± 11.3 kg) performed simulated falls and physical activities of daily life. Every participant performed four kinds of simulated falls and seven other types of PAs. Every simulated fall and PA were repeated three times by each of the participants. The protocol of this study was approved by the Research Ethics Committee of the Chang Gung Medical Foundation (IRB#201802118B0) in accordance with the Helsinki Declaration. All of the participants provided written informed consent.

The participants were protected by a thick mattress while undertaking the simulated falls. The four kinds of simulated falls were defined as follows:(1)A backward fall during stand-to-sit.(2)A forward fall, a lateral fall toward the left side, and a lateral fall toward the right side during walking.(3)A forward fall, a lateral fall toward the left side, and a lateral fall toward the right side from standing.(4)A forward fall, a lateral fall toward the left side, and a lateral fall toward the right side during sit-to-stand.

The seven types of PAs were defined as follows:(1)A single jump and a continuous jump.(2)A forward jog at an expected speed of 1.6 m/s and jogging in place with an expected time interval of 1.5 s.(3)Forward walking at two expected speeds (0.8 and 1.3 m/s), walking backward, and walking in place with an expected time interval of 2 s on level ground.(4)Walking downstairs.(5)Walking upstairs.(6)Sitting straight and sitting with legs stretched out.(7)Standing still, standing with hands raising, and standing with body swing in anteroposterior/left-right/up-down direction.

### 2.2. Sensing and Recording System

An ARM Cortex-M4 microcontroller (M451RG6AE, Nuvoton Tech. Corp., Hsinchu, Taiwan) received the digital data from an IMU (LSM9DS1, STMicroelectronics, Geneva, Switzerland) with a full-scale, ranged ±4 g tri-axial accelerometer, a full-scale, ranged ±500 degrees/s (dps) ±12 Gauss tri-axial magnetometer, via a serial peripheral interface bus at a sampling rate of 100 Hz. As shown in Figure 1, a customized insole with 11 FSRs (UNEO Incorporated, New Taipei City, Taiwan) was used to capture the foot pressures at the big toe, little toe, metatarsus (medial, middle, lateral), arches (medial, lateral), fore heels (medial, lateral), and heels (medial, lateral). The FSRs were constructed of a resistance-type piezo-resistive polymer composite made using processing and printing-based micromachining technology. Each FSR had a sensing range of 1 to 5 kg/cm^2^ and was individually calibrated using elastic-film pressurization to reduce the resistance variance between the sensors. The microcontroller digitized the transformed voltages from these FSRs through a built-in 12-bit analog-to-digital converter at a sampling rate of 100 Hz. All of the acquired samples were wirelessly transmitted to a notebook computer through a BLE 4.2 Bluetooth module (JDY-18, Shenzhen Innovation Technology, Shenzhen, China). The customized insole had a height of 260 mm, a metatarsus width of 850 mm, a heel width of 550 mm, and a 0.63 mm thickness. Therefore, we just included participants whose foot size could match the size of the customized insole as much as possible.

The sensing devices were fixed at the lateral aspect or insole of each shoe in parallel to measure the foot inertial data and foot pressures, respectively. A graphical user interface developed in the PyQt Designer (Riverbank Computing, Dorchester, UK) collected these data in parallel with a video recording from a Kinect V2 camera (Microsoft Corp., Redmond, WA, USA) at a rate of 30 frames/s.

### 2.3. Fall and PA Detection Network

A network to detect falls and various types of PAs was constructed based on the CNN and RNN, which are constructed using a deep residual network (DRN) and a bidirectional long short-term memory (LSTM) network. The inputs can be foot inertial data and/or three parameterized data from a single foot or both feet in a 3-s window. The inputs can also be foot pressures and/or the center of pressure (CoP) from a single foot or both feet. The combinations of these data and parameters are also used.

The foot inertial data and three parameterized data are described as follows:(1)Tri-axial accelerations: *a_x_*(*i*), *a_y_*(*i*), *a_z_*(*i*)(2)Tri-axial angular velocities: *ω_x_*(*i*), *ω_y_*(*i*), *ω_z_*(*i*)(3)Acceleration amplitude (AM) defined as the square root of the sum of tri-axial accelerations [32]:
$$\mathrm{AM}\left(i\right)=\sqrt{a_{x}^{2}\left(i\right)+a_{y}^{2}\left(i\right)+a_{z}^{2}\left(i\right)}$$(4)Acceleration cubic-product-root magnitude (ACM) defined as the cube root of the product of tri-axial absolute accelerations [44]:
$$\mathrm{ACM}\left(i\right)=\sqrt[{3}]{\left|a_{x}\left(i\right)\times a_{y}\left(i\right)\times a_{z}\left(i\right)\right|}$$(5)Angular velocity cubic-product-root magnitude (AVCM) defined as the cube root of the product of tri-axial absolute angular velocities [44]:
$$\mathrm{AVCM}\left(i\right)=\sqrt[{3}]{\left|\omega_{x}\left(i\right)\times \omega_{y}\left(i\right)\times \omega_{z}\left(i\right)\right|}$$

Figure 2 shows the tri-axial accelerations, tri-axial angular velocities, and the corresponding inertial parameters (AM, ACM, and AVCM) while a participant was undertaking a forward fall, a jump, a forward walk, and a forward jog. Both falling and jumping produced abrupt changes in accelerations and angular velocities. Two distinct acceleration peaks were generated particularly while jumping off ground and back to ground. Both walking and jogging produced repeated changes in accelerations and angular velocities.

The foot pressures and their CoP are described as follows:(1)Eleven individual foot pressures: *fp*_1_(*i*), *fp*_2_(*i*), …, *fp*_11_(*i*)(2)CoP in horizontal direction (CoPx)(3)CoP in vertical direction (CoPy)

CoPx and CoPy of the left or right foot are, respectively, defined as the sum of the products of the eleven individual foot pressures and their x and y positions divided by the sum of the eleven foot pressures:$$\mathrm{CoPx}\left(i\right)=\frac{\sum (fp_{n}\left(i\right)*x_{n})}{\sum fp_{n}\left(i\right)}$$
$$\mathrm{CoPy}\left(i\right)=\frac{\sum (fp_{n}\left(i\right)*y_{n})}{\sum fp_{n}\left(i\right)}$$
where *x_n_* and *y_n_* indicate the location of the centroid of the *n*-th FSR relative to the local reference frame.

The CoP of both feet is defined as the weighted sum of the left-side CoP and right-side CoP. When non-ground contact is detected, CoP is set to the center of a single foot or the center of both feet.

Figure 3 shows the foot pressures at the metatarsus and heel areas, and the CoP trajectories during the same activity events. The foot pressures were close to zero after the fall incidence. There were also two distinct peak pressures at the metatarsus area while jumping off ground and back to ground. The foot pressures at the metatarsus and heel areas were interlaced while walking or jogging, which were linked to toe-off and heel-strike. Moreover, various CoP trajectory patterns were presented in falling, jumping, walking, and jogging. In particular, both walking and jogging produced butterfly patterns.

The inputs, therefore, have a size of M (channels) × 300 (points). As shown in Figure 4, a DRN is constructed by a stack of three residual units (RUs). Each RU is composed of six convolutional layers and a skip layer. In each convolutional layer, an equivalent number of 1-D filters are used to capture the temporal patterns of the inputs individually, preserving the temporal dimensions (stride 1, same padding); the temporal patterns are then summed along the channel dimension and outputted through a ReLU activation function. Each convolutional layer has multiple sets of 1-D filters and outputs one feature channel per set. A max-pooling layer subsamples the outputs of the last convolutional layer to halve the dimensionality of the features. The skip layer adds the inputs of the RU directly to the halved output features through a convolution layer with a stride of 2 and the right number of output channels. The detail of the deep residual network is listed in Table 3. The kernel size is the same within each RU but differs across RUs.

The benefits of the residual network have been shown through its easier optimization and the accuracy obtained from the increased depth [66]. The purpose of using three RUs is to extract the low-, middle-, and high-level features sequentially from the signals. The kernel size of these three RUs was 11, 9, and 5, respectively. With the increased level, the number of feature maps is increased, and the dimensionality of the features is halved.

The output of the deep residual network can be viewed as a sequence of feature vectors (38 vectors with vector size 108) such that it is inputted to a bidirectional LSTM network. The current input vector and previous short-term state vector are fed into a fully connected layer on which the LSTM cell generates a short-term vector and a long-term vector (each has a vector size of 64) to the next state. After the bidirectional LSTM network has been traversed in a forward direction, it is then traversed backward. The short-term vector of the last state is used as an input to a terminal fully connected network, which contains one hidden layer of 32 neurons and one output layer of 8 neurons corresponding to a fall and 7 types of PAs. The activation function in the hidden layer is chosen as the ReLU function, where the output layer uses a softmax function to represent a categorical probability distribution.

A batch containing 96 randomly chosen instances was inputted into the deep neural network. The corresponding output vectors were calculated via forward propagation. Therefore, the cross-entropy loss was computed based on the forward output vectors and the one-hot target vectors. A backpropagation of the errors was subsequently applied to update the network weights using the Adam optimization algorithm, which estimates the updates using a running average of the first and second moment of the gradient [67].

The settings of the hyper-parameters with a brief description of the forward processing and consideration are summarized in the following. These settings were determined through various tests on our dataset and achieved the optimal prediction performance.

(1)A batch size of 96 to achieve a balance between the robustness of stochastic gradient descent and the efficiency of batch gradient descent.(2)Three RUs to extract low-, middle-, and high-level features from signals: 1D convolutions with a kernel size of 11 in the first RU to capture the temporal patterns of signals (~110 ms), 9 in the second RU to capture the low-level temporal features, and 5 in the third RU to aggregate the middle-level features.(3)36, 72, and 108 feature maps to generate sufficient low-, middle- and high-level features.(4)Two sequential 64-unit LSTM cells to cluster the high-level features at various time points.(5)A fully connected network containing a hidden layer of 32 neurons to aggregate the last-state short-term vectors and an output layer of 8 neurons to classify falls and 7 types of PAs.

The proposed detection network was implemented in the Visual Studio Code Ver. 1.57.0 (Microsoft Corp., Redmond, WA, USA) using Python Ver. 3.8.10 (Python Software Foundation, Wilmington, DE, USA) and the Keras API of Tensorflow 2.0 (Google Brain, Mountain View, CA, USA). The networks were trained, validated, and tested in a server computer with an 8-core CPU (Intel^®^ CoreTM i9-9900K, Santa Clara, CA, USA) and a 24-GB GPU (Titan RTX, Nvidia Corp., Santa Clara, CA, US) using CUDA technology.

### 2.4. Network Training and Testing

In order to generate the data for network training and testing, foot pressures and foot inertial data with a duration of 4.5 s that covered a fall or a PA were selected as a basis for data augmentation.

(1)Several 3-s fall segments were obtained by choosing various onset times at 0, 0.5, 1, and 1.5 s and four additional multiples by randomly shifting the onset time within the three intervals (0–0.5 s, 0.5–1 s, 1–1.5 s). Therefore, a total of 3 × 10 × 16 segments (10 falls, each fall had 3 trials, 16-time data augmentation) were obtained from each participant and labeled as falls (class 0).(2)Several 3-s PA segments were obtained by choosing various onset times at 0, 0.5, 1, and 1.5 s and four additional multiples by randomly shifting the onset time within three intervals (0–0.5 s, 0.5–1 s, 1–1.5 s). The number of the augmented segments depended on the type of PA such that a total of 3 × 160 segments (each movement had 3 trials) were obtained from each PA for each participant and labeled as a specific PA (class 1 to class 7). Therefore, an evenly balanced dataset between falls and various types of PA was generated.

The dataset, therefore, contained 32 × 3840 instances (32 participants, 480 instances per class in 8 classes related to fall and seven types of PAs). Each instance was constituted by 34 channels × 300 samples (3-s segment with 3-axial accelerations, 3-axial angular velocities, and 11 foot pressures on each shoe).

The foot pressures were normalized by the sum of the eleven standing foot pressures collected from each participant. The inertial data, the normalized foot pressures, and their parameterized data were separately standardized between 0 and 1 across participants.

Leave-one-out cross-validation (LOOCV) was used to validate the neural network. The data of one participant were chosen to validate the network determined by the data from the remaining participants (training set). The LOOCV was repeated to allow each participant’s combination of falls and PAs to be the validation data once.

A total of 50 iterations were applied to train the networks using an Adam optimization algorithm based on the cross-entropy loss function. A fraction of the hidden units in the terminal fully connected network was randomly dropped at every iteration with a probability of 0.5 to force the network to learn general and robust patterns from the data to prevent overfitting.

All of the validation data were fed forward to the trained model. The output values were then compared with the labeled values. The recall, precision, and F1-score for the fall or each PA were calculated based on all LOOCVs. The recall was defined as the percentage of true positives (fall or PA was correctly classified) among all validation data, while the precision was defined as the number of true positives divided by the number of true positives and false positives (the misclassified fall or the misclassified PA). The F1-score was computed by 2 × recall × precision/(recall + precision).

## 3. Results

Table 4 lists the results of the fall detection and various types of PA based on foot pressures, CoP, and their combination obtained from both feet. We used F1-score as an index to evaluate the detection performance. Falls were well detected, particularly when using the combination of foot pressures and CoP. Static PAs (sitting, standing) were also best detected using this combination. However, the incorporation of foot pressures to detect walking upstairs and downstairs was limited. The poor detection was improved when using CoP alone. Similarly, other dynamic PAs (jumping, jogging, level walking) were best detected using CoP only.

Table 5 lists the results of the detections of the falls and various types of PA based on the inertial data, inertial parameters, and their combination obtained from both feet. The use of inertial data provided better detections (higher F1-scores) for falls and most PAs except jogging. The combination of inertial data and inertial parameters improved the detection of sitting, jumping, jogging, and walking downstairs. Compared to the foot pressure-based measurements (Table 2), the inertial-based measurements detected fall and dynamic PAs better but had worse detection for static PAs (sitting and standing).

As listed in Table 6, falls and various types of PA were well detected when the foot pressure-based measurements and inertial-based measurements were used in combination.

Table 7 summarizes the detection of particular performances based on foot pressures, CoP, and the inertial-based measures obtained from the left foot, right foot, or both feet. The application of right foot data performed best when detecting falls and various types of PA.

Table 8 lists the results of the classifications of falls and various types of PAs using data from both feet. The number of each true class is 480 for each participant. The listed values are the average misclassifications over 32 participants when the inertial data and inertial parameters (1st row), CoP (2nd row) and inertial data, inertial parameters, foot pressures, and CoP (3rd row) are used, respectively. Overall, 14.7% (70.63/480) and 19.8% (94.84/480) of the results were misclassifications between sitting and standing, which were found by using the inertial data and inertial parameters. The misclassifications were improved when the inertial data, inertial parameters, CoP, and foot pressures were used in combination. The use of inertial data and inertial parameters resulted in more than 2% misclassifications from jogging, upstairs walking to jumping. The use of CoP alone led to more than 2% misclassifications from falling to jumping, from jumping to falling or standing, from jogging to downstairs walking, from upstairs walking to falling, and among level walking, upstairs walking and downstairs walking. These misclassifications were improved when the inertial data, inertial parameters, CoP, and foot pressures were used in combination.

Table 9 lists the computational complexity of the proposed deep neural model with various inputs. The number of weights and the number of computations slightly increased with the number of input features. The time for model training in each leave-one-out cross-validation is 30 min. A total of 16 h was needed to complete 32 cross-validations, which allowed every participant’s data to test the model trained by the other 31 participants’ data.

## 4. Discussion

Falls and PAs are detected or classified based on different rationales. Falls are usually characterized by impact-on-ground and post-impact lying such that several parameter-based methods have been developed to capture these characteristics, whereas PAs have a variety of kinematic patterns depending on the included PA types. Deep-learning neural networks provide a way to extract multiple varieties of PA-related features through network training on labeled data. In this study, we detected falls and various types of PAs based on a deep-learning architecture that is constituted using three networks. First, several multi-layer 1-D convolutional networks extract the low- to high-level features. Second, a skip connection feeds the raw data or features to the output of multiple convolutional layers to reduce feature degradation. Third, a bidirectional LSTM network clusters the output features at various time points to capture the sequential pattern of a fall or PA.

Inertial sensing on the chest, waist, or wrist is the commonly used measurement for fall detection in the literature. The measurement based on foot pressures and foot inertial data provides an alternative approach to distinguish falls from PA. One study built empirical rules based on the foot’s acceleration magnitude (AM) and inactivity duration on the ground [64], but the rules were hand-crafted and only validated on six participants. Another study employed a decision tree to determine the threshold of the waist’s AM or the CoP of both feet for fall detection, which led to a better performance (area under receiver operating characteristic curve) using the waist’s AM and an improved performance when both features were used [65]. However, the performance using foot inertial data was not reported. In this study, we investigated the availability of deep neural networks based on foot-pressure-based or/and foot-inertial-based measures on fall detection. The F1-score was used because it was an overall index between the sensitivity of fall detection and the ratio of correctness in the detected falls. Our result showed a recall of 0.971, a precision of 0.943, and an F1-score of 0.980 in fall detection based on foot pressures and CoP, which supported the existence of fall-related information from foot pressures in accordance with different temporal foot pressure distributions during fall relative to PAs [65]. In addition, using foot inertial data provided a recall of 0.999, a precision of 0.998, and an F1-score of 0.998 for fall detection. The excellent performance of foot inertial data could be attributed to the significant changes in both acceleration and angular velocities during falls, particularly during ground impact.

In this study, we extended fall detection to various types of PAs and applied a deep-learning model to handle these multiple classifications. Our results showed that the inertial-based measures outperformed the foot-pressure-based measures on the detection of dynamic PAs, i.e., the inertial-based measures presented more distinct patterns among jumping, jogging, level walking, upstairs walking, and downstairs walking than the foot-pressure-based measures. On the contrary, using inertial-based measures to detect static PAs was limited, whereas using the foot-pressure-based measures worked well. Therefore, the combination of foot-pressure-based and inertial-based measures was suggested because it achieved a good performance in detecting both static PAs and dynamic PAs.

It is not easy to compare the efficiency of the detection model among various studies because there are different PA types in different datasets. Nevertheless, our model also produced a similar trend in the F1-score between PA classification and fall detection, as in the previous works listed in Table 1 and Table 2; that is, the accuracy of fall detection was higher than those of PA detection. We speculate that falls create quite different temporal patterns in contrast to PAs, whereas the movements of some PAs are somewhat similar, e.g., among level, upstairs and downstairs walks. Therefore, misclassifications among these walks were higher than their false negatives as other PAs or false positives from other PAs, as shown in the confusion matrix (Table 8).

A limitation of this study is that all of the data were collected from young, healthy participants. However, it is not rational or ethical to invite elderly subjects to perform simulated falls for safety reasons. In practical applications, the detection model can be pre-trained based on a large healthy dataset; once a smaller elderly dataset is obtained, transfer learning can be applied to fine-tune the model.

In this study, the performance of fall and PA detections based on a single foot was similar or even higher on the right foot than that based on both feet. In fact, a single-foot measurement is more readily applicable in daily life since a stand-alone detection device can be mounted on one shoe without the need for data transmission between the left and right sides. In cases such as post-stroke hemiplegia or Parkinson’s disease, the application of the device on the unaffected side would be especially easier. We speculate that the unaffected side would produce a pattern closer to younger subjects if compared to the affected side. Similarly, transfer learning can be employed to fine-tune the trained model.

CoP is commonly given by the sum of the products of individual foot pressures and their x and y positions divided by the sum of the foot pressures. This calculation yields a high-precision estimation while using insoles with high-density foot-pressure sensing. It is also applied to the CoP estimation using instrumental insoles with eight foot-pressure sensors on each side [68], 10 sensors on both feet [65], the optimal placement of eight sensors on the left foot [69] or the optimal 13 sensors on both feet [70]. In order to improve the precision of the CoP estimation by the reduced number of foot-pressure sensing, a linear regression calibration [70] or a feed-forward neural network [68] was proposed. Our work focused on fall detection and PA classification based on the customized insole with 11 FSR sensors and demonstrated that the proposed deep neural network could learn discernible features from the crudely estimated CoP for the detection of falls and various types of PAs. Further study is needed to clarify whether calibrated CoP can improve detection.

Healthcare monitoring based on wearable devices has received considerable interest in the management of physiology and psychology. It is worth noting that the measured physiological and behavioral information can be gathered via Internet of Things (IoT) technology, and the increased amount of the gathered information requires further processing to broaden its application through deep learning techniques [71,72]. Wearable IoT sensors provide a solution for the objective remote monitoring of real-life ADL and real fall events for activity-level evaluation, fall prevention, and risk assessment in the elderly and subjects with dementia, Parkinson’s disease, cardiovascular disease, and frailty [73].

Detection methods that rely on the inertial sensing of the human body and limbs are light, easy to wear, and low-cost and can be practically implemented in the real world, but the applications may be limited when the subjects are not willing to or forget to wear these devices. The proposed footwear approach is beneficial in that the sensors can be embedded in shoes, which are commonly worn during off-bed activities, particularly outdoor activities. The cost and complexity of the proposed footwear device can be further reduced by considering the use of one-foot measurements with a smaller number of foot pressures. Further study is needed to investigate the effect of dimensionality reduction on detection performance in future work. In addition to the proposed fall and PA detection, foot-pressure monitoring also provides plantar-pressure information, which is used as a biomechanical assessment for body balance and ergonomics posture during static or dynamic gait [74].

Two potential problems may affect the detection performance while applying the trained deep-learning model in a real scenario. The first problem is the existence of overfitting at the training stage. The trained model cannot efficiently predict the results on the new data when there exists variance with the trained data. Several works have addressed this problem and utilized some approaches to avoid overfitting. For instance, a penalty term is added to the loss function to optimize the boundary of features [75,76]; data augmentation is used to allow the model to more accurately catch different data structures [77]; dropout is applied to prevent neurons from co-adapting too much [78]; early stopping is adopted to reduce unnecessary computing [79]. The above strategies allow the model reduce the focus on some rare specific features during the training phase, thereby keeping the balance and flexibility of the model. In our work, we expanded the collected data by sixteen multiples to enable the model to gather more detail across various situations. Considering the computational complexity of the proposed CNN + LSTM model (Table 9), we used the 50% dropout of neurons randomly at each training epoch to avoid overfitting as much as possible and reduced the computational complexity of model training as well. Additionally, early stopping was applied when the validation loss increased.

The second problem is the effect of noise on the detection performance. At present, the data collected by the inertial sensors and FSRs are not proven to be perfect. Every type of measured signal (accelerations, angular velocities, and foot pressures) in our device possibly has interferences from movement disturbances. A study applied the Dempster–Shafer theory [80] to conduct decision-level image fusion to improve the crack detection accuracy with high robustness to the noise effects [81]. Therefore, it is possible to incorporate the Dempster–Shafer algorithm to fuse data from different sensors [82] for the detection of falls and PAs in our deep neural model in future work.

## 5. Conclusions

We focused on the employment of foot-pressure-based and foot inertial-based measures to detect falls and various types of PAs using a deep neural network that used CNN to extract discernible features and RNN to cluster the features at various time points. Foot-pressure-based measures, as well as foot inertial-based measures, performed well in fall detection. Foot inertial-based measures led to better performance in detecting dynamic PAs (jumping, jogging, walking), while the foot-pressure-based measures yielded better performance in detecting static PAs (sitting, standing). The combination of foot-pressure-based and foot-inertial-based measures allowed for the detection of both static PAs and dynamic PAs. Although the capability of a deep neural network based on foot pressures and foot inertial sensing to detect falls, static PAs, and dynamic PAs was demonstrated based on the collected data from young participants, the model can be fine-tuned by transfer learning for practical application to the elderly. Moreover, the investigation of one-foot measurement with a fewer number of foot pressures can be conducted to reduce the complexity of the proposed footwear approach in the future.

## Figures and Tables

**Figure 1 sensors-23-00495-f001:**
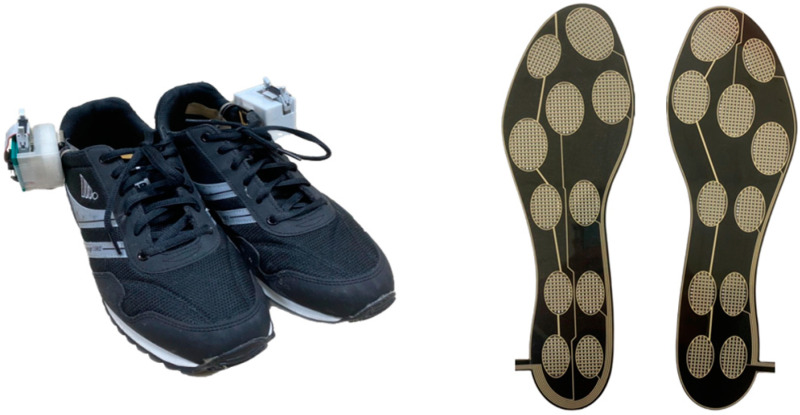
Instrumental shoes with inertial sensing and foot pressure measurements at the big toe, little toe, metatarsus (medial, middle, lateral), arches (medial, lateral), fore heels (medial, lateral), and heels (medial, lateral).

**Figure 2 sensors-23-00495-f002:**
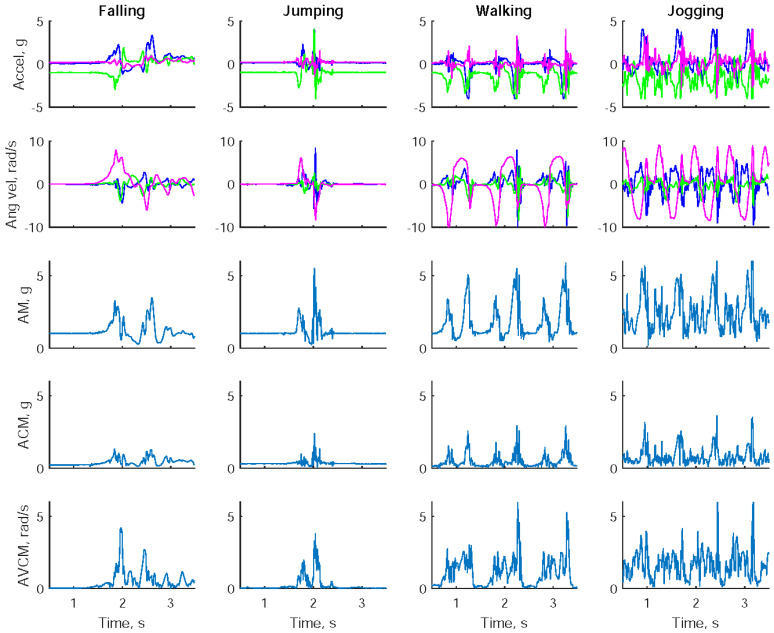
Tri-axial accelerations, tri-axial angular velocities, and the corresponding inertial parameters including acceleration amplitude (AM), acceleration cubic-product-root magnitude (ACM), and angular velocity cubic-product-root magnitude (AVCM) while a subject was undertaking a forward fall, a jump, a forward walk, and a forward jog.

**Figure 3 sensors-23-00495-f003:**
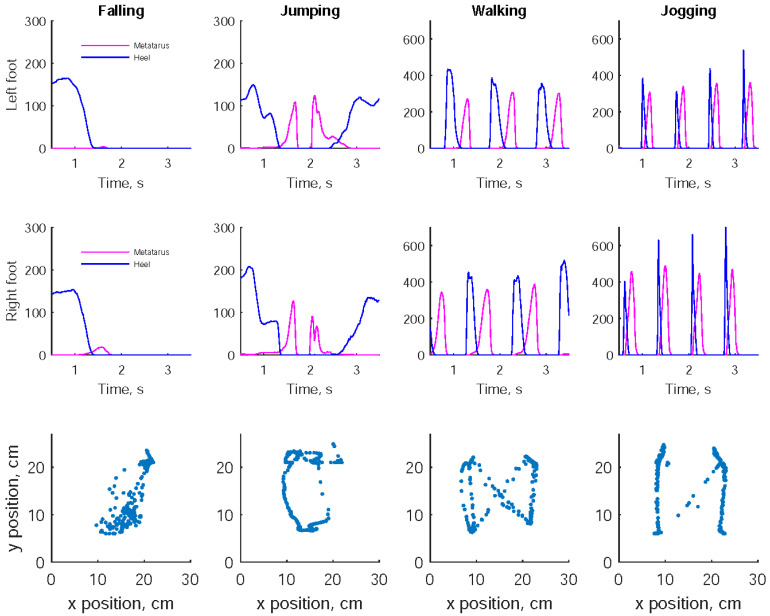
Relative foot pressures at the metatarsus, heel areas, and center of plantar pressure while a subject was undertaking a forward fall, a jump, a forward walk, and a forward jog.

**Figure 4 sensors-23-00495-f004:**
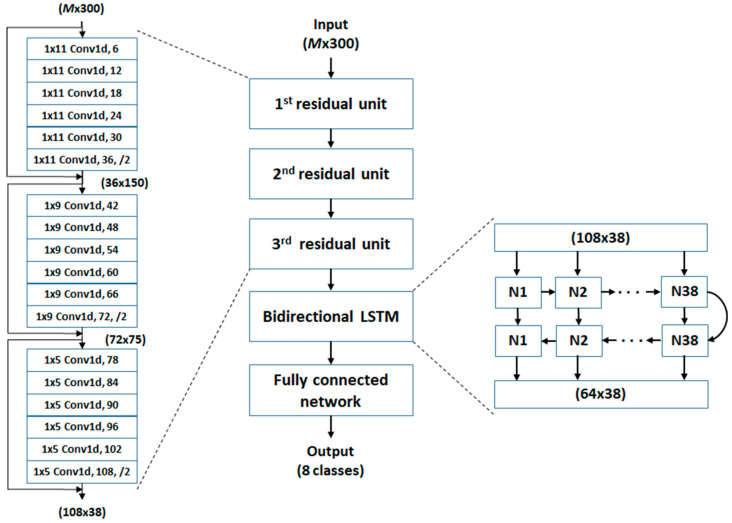
Deep neural network for detecting fall and 7 types of physical activities based on foot pressures and foot inertial data (*M* channels).

**Table 1 sensors-23-00495-t001:** Methods for physical activity (PA) classifications.

Authors	Dataset/ Subject No.	Sensor	Placement	Feature	Model	PA Type	Accuracy
Kwapisz et al. [25]	WISDM/29	A	leg	statistical features	MLP	jog, stand, sit, walk, stairs	0.917
Micucci et al. [26]	UniMiB-SHAR/30	A	thigh	signals	kNN	jump, jog, lay down, stand up, sit down, walk, stairs	0.830
Anguita et al. [31]	UCI HAR/30	A, G	waist	time and freq features	SVM	lay, stand, sit, walk, stairs	F1 = 0.960
Sztyler & Stuckenschmidt [27]	Realworld HAR/15	A	chest, head, waist, arm, thigh, shin	time and freq features	RF	jump, jog, lay, stand, sit, walk, stairs	F1 = 0.890
Chavarriaga et al. [29]	12	A, G	trunk, arm, knee, foot	principal components	kNN	lay, stand, sit, walk	F1 = 0.850
Garcia-Gonzalez et al. [30]	15	A, G	long-term smartphone	statistical features	SVM	inactive, active, walk, drive	0.744
Moufawad el Achkar et al. [63]	10	A, G, B, FP	trunk, thigh, foot		DT	stand, sit, walk, stairs, uphill, downhill	0.970
Qi et al. [48]	20	A, G, M	waist	principal components	CNN; LSTM	jump, jog, lay, stand, walk, stairs	0.953; 0.964
Almaslukh et al. [49]	Realworld HAR/15 [27]	A	chest, head, waist, arm, thigh, shin	signals	CNN	jump, jog, lay, stand, sit, walk, stairs	F1 = 0.860
Avilés-Cruz et al. [50]	UCI HAR/30 [31]	A, G	waist	signals	CNN	lay, stand, sit, walk, stairs	1
Russell et al. [51]	-	A	chest	signals	CNN	lay, sit, walk, climb	0.982
Huang et al. [52]	WISDM/36 [25]	A	leg	signals	CNN	jog, stand, sit, walk, stairs	F1 = 0.940
Chen et al. [62]	30	A, G	foot	signals	CNN	Uphill, downhill, slope, walk, stairs	0.877
Murad & Pyun [55]	USC HAD/14 [28]	A, G	hip	signals	RNN	jump, jog, lay, stand, sit, walk, stairs	F1 = 0.970
Fridriksdottir & Bonomi [59]	20	A	trunk	signals	CNN + LSTM	stand, sit, walk, stairs	F1 = 0.946
Ankita [60]	UCI HAR/30 [31]	A, G	waist	time and freq features	CNN + LSTM	lay, stand, sit, walk, stairs	0.979

Methods in the first block use the hand-crafted features that are obtained from foot pressures (FP), barometer (B), inertial signals by accelerometer (A), gyroscope (G), and magnetometer (M). Multilayer perceptron (MLP), support vector machine (SVM), k-nearest neighbors (kNN), decision tree (DT), and random forest (RF) are therefore used to classify PA based on the hand-crafted features or signals. Methods in the second block use convolutional neural networks (CNNs), recurrent neural networks (RNNs), long short-term memory (LSTM), and their combination (CNN + LSTM) to learn discriminant features form inertial signals and classify PA.

**Table 2 sensors-23-00495-t002:** Methods for fall detections.

Authors	Dataset/ Subject No.	Sensor	Placement	Feature	Model	Activity of Daily Life	Accuracy
Karantonis et al. [32]	6	A	waist	AM, posture	empirical	walk, PT	0.908
Bourke et al. [33]	20	A	chest; thigh	AM	Empirical	walk, PT	1 ^†^; 0.913 ^†^
Kangas et al. [38]	3	A	waist, wrist, head	AM, posture	empirical	walk, stairs, PT	0.990 ^†^
Chao et al. [39]	7	A	chest; thigh	AC, posture	empirical	walk, squat, PT	0.991 ^†^; 0.996^†^
Bourke et al. [40]	20	A	waist	AC, VV, posture	empirical	walk, PT	1
Sucerquia et al. [36]	SisFall/38	A	waist	AM	empirical	jump, jog, walk, PT	0.978
Medrano et al. [35]	DFNAPAS/10	A	pocket	AM	SVM	real-life ADL	0.939 ^†^
Ojetola et al. [34]	Cognent/32	A, G	chest	AM	empirical	stand, lay, sit, walk, crouch, PT	F1 = 0.940
Nyan et al. [42]	10	G	chest, waist	angular velocity	empirical	walk, PT	0.987 ^†^
Bourke et al. [43]	20	G	chest	angular velocity and acceleration	empirical	walk, PT	1
Casilari et al. [41]	UMAFall/17	A, G	chest, waist, thigh, wrist	AM, DA, posture	empirical	jump, walk, stairs, PT	0.886
Wang et al. [44]	15; Cognent/32 [34]; UMAFall/17 [41]	A, G	chest	AM, ACM, AVCM	empirical	jump, walk, PT; crouch, walk, PT; jump, walk, stairs, PT	0.991 ^†^; 0.973 ^†^; 0.896 ^†^
Zitouni et al. [64]	6	A	foot	AM, DA, posture	empirical	stand, lay, sit, walk, sport, PT	0.966 ^†^
Lee et al. [65]	9	A, FP	foot	AM, DA, CoP	DT	stand, lay, sit, walk, PT	0.951
Santos et al. [45]	URFD/5 [17]; SmartFall/7 [53]	A, G	hip; wrist	signals	CNN	lay, sit, walk, crouch; jog, sit, hand move	0.999; 0.999
Ribeiro et al. [46]	12	A, G, M	back, thigh, foot	principal components	CNN	walk	0.927
Casilari et al. [47]	SisFall/38 [36]; UMAFall/17 [41]	A	waist	AM, accelerations	CNN	jump, jog, walk, PT;jump, walk, stairs, PT	0.988; 0.821
Mauldin et al. [53]	SmartFall/7	A	wrist	signals	RNN	jog, sit, hand move	0.850
Luna-Perejón et al. [54]	SisFall/38 [36]	A	waist	signals	RNN	jump, jog, walk, PT	0.967
Theodoridis et al. [58]	URFD/5 [17]	A	hip	signals	CNN + LSTM	lay, sit, walk, crouch	0.986
Delgado-Escaño et al. [56]	DFNAPAS/10 [35]; SisFall/38 [36]; UniMiB-SHAR/30 [26]	A	pocket; waist; thigh	signals	CNN + LSTM	real-life ADL; jump, jog, walk, PT; jump, jog, walk, stairs, PT	0.997; 0.972; 0.873
Liu [57]	SisFall/38 [36] MobiFall/24 [37]	A, G	waist	signals	CNN + LSTM	jump, jog, walk, stairs	0.992

Methods in the first block use hand-crafted features that are obtained from foot pressures (FP) and inertial signals by accelerometer (A), gyroscope (G), and magnetometer (M). The features including acceleration amplitude (AM), acceleration cross-product (AC), differential acceleration (DA), vertical velocity (VV), acceleration cubic-product-root magnitude (ACM), angular velocity cubic-product-root magnitude (AVCM), angular velocity, angular acceleration, and posture are therefore used to distinguish falls from activities of daily life (ADL) based on empirical rules or a support vector machine (SVM) or decision tree (DT). Methods in the second block use convolutional neural networks (CNNs), recurrent neural networks (RNNs), long short-term memory (LSTM), and their combination (CNN + LSTM) to learn discriminant features form inertial signals and distinguish falls from ADL. ^†^ indicates the accuracy (acc) is calculated by $\sqrt{\mathrm{sensitivity}\times \mathrm{specificity}}$.

**Table 3 sensors-23-00495-t003:** Deep residual network architecture.

Residual Unit (RU)/Layer	The 1st RU Kernel Size 11, Stride 1			The 2nd RU Kernel Size 9, Stride 1			The 3rd RU Kernel Size 5, Stride 1		
	Input Channel	Output Channel	Size	Input Channel	Output Channel	Size	Input Channel	Output Channel	Size
Conv1d	*M*	6	300	36	42	150	72	78	75
Conv1d	6	12	300	42	48	150	78	84	75
Conv1d	12	18	300	48	54	150	84	90	75
Conv1d	18	24	300	54	60	150	90	96	75
Conv1d	24	30	300	60	66	150	96	102	75
Conv1d	30	36	300	66	72	150	102	108	75
Max pooling kernel size 2 stride 2	36	36	150	72	72	75	108	108	38
Skip layer kernel size 1 stride 2	*M*	36	150	36	72	75	72	108	38

**Table 4 sensors-23-00495-t004:** Results on the detections of fall and physical activities based on foot pressures and center of pressure obtained from both feet.

	Fall	Sit	Stand	Jump	Jog	LW	US	DS	Macro
Foot pressures									
Recall Precision F1-score	0.935 0.973 0.950	0.992 0.963 0.973	0.850 0.922 0.875	0.837 0.825 0.812	0.898 0.846 0.853	0.755 0.819 0.764	0.644 0.569 0.555	0.622 0.465 0.487	0.817 0.798 0.784
Center of pressure									
Recall Precision F1-score	0.953 0.888 0.913	0.856 0.937 0.877	0.923 0.869 0.885	0.876 0.951 0.902	0.945 0.977 0.958	0.922 0.882 0.887	0.762 0.877 0.783	0.848 0.893 0.844	0.886 0.909 0.881
Foot pressures + center of pressure									
Recall Precision F1-score	0.971 0.943 0.980	0.976 0.989 0.980	0.946 0.906 0.921	0.891 0.924 0.897	0.900 0.901 0.881	0.852 0.869 0.841	0.602 0.638 0.580	0.574 0.671 0.592	0.839 0.855 0.831

Macro average used to calculate recall, precision, and F1-score of multiple classes; LW, level walking; US, upstairs walking; DS, downstairs walking.

**Table 5 sensors-23-00495-t005:** Results on the detections of fall and physical activities based on inertial data and parameters obtained from both feet.

	Fall	Sit	Stand	Jump	Jog	LW	US	DS	Macro
Inertial data									
Recall Precision F1-score	0.999 0.998 0.998	0.724 0.887 0.770	0.862 0.781 0.800	0.976 0.957 0.963	0.915 0.962 0.930	0.974 0.965 0.968	0.972 0.974 0.968	0.987 0.994 0.990	0.926 0.940 0.923
Inertial parameters									
Recall Precision F1-score	0.995 0.969 0.980	0.769 0.790 0.759	0.774 0.788 0.767	0.941 0.956 0.940	0.932 0.958 0.938	0.963 0.940 0.946	0.937 0.990 0.954	0.961 0.973 0.958	0.909 0.921 0.905
Inertial data + inertial parameters									
Recall Precision F1-score	0.994 0.988 0.990	0.854 0.819 0.804	0.742 0.887 0.762	0.968 0.968 0.966	0.943 0.958 0.946	0.962 0.964 0.961	0.964 0.973 0.964	0.990 0.995 0.993	0.927 0.964 0.923

Macro average used to calculate recall, precision, and F1-score of multiple classes; LW, level walking; US, upstairs walking; DS, downstairs walking.

**Table 6 sensors-23-00495-t006:** Results on the detections of fall and physical activities based on Inertial data and parameters obtained from both feet.

	Fall	Sit	Stand	Jump	Jog	LW	US	DS	Macro
Foot pressures + center of pressure + inertial data + inertial parameters									
Recall Precision F1-score	0.996 0.997 0.996	0.987 0.995 0.990	0.990 0.979 0.983	0.981 0.965 0.971	0.924 0.953 0.931	0.917 0.957 0.960	0.919 0.948 0.924	0.979 0.987 0.983	0.968 0.973 0.967
Center of pressure + inertial data + inertial parameters									
Recall Precision F1-score	0.998 0.991 0.994	0.904 0.961 0.924	0.958 0.920 0.934	0.991 0.980 0.985	0.956 0.957 0.953	0.966 0.960 0.961	0.926 0.964 0.938	0.981 0.991 0.985	0.960 0.966 0.959

Macro average used to calculate recall, precision, and F1-score of multiple classes; LW, level walking; US, upstairs walking; DS, downstairs walking.

**Table 7 sensors-23-00495-t007:** Results on the detections of fall and physical activities based on foot pressures, center of pressure, inertial data and inertial parameters obtained from a single foot or both feet.

	Fall	Sit	Stand	Jump	Jog	LW	US	DS	Macro
Left foot									
Recall Precision F1-score	0.998 0.995 0.996	0.969 0.988 0.972	0.978 0.970 0.970	0.922 0.929 0.919	0.883 0.926 0.894	0.961 0.935 0.945	0.946 0.963 0.950	0.981 0.979 0.979	0.955 0.961 0.953
Right foot									
Recall Precision F1-score	0.998 0.996 0.997	0.977 0.998 0.985	0.990 0.971 0.979	0.977 0.951 0.960	0.920 0.967 0.935	0.972 0.962 0.966	0.952 0.963 0.951	0.982 0.994 0.988	0.971 0.975 0.970
Double feet									
Recall Precision F1-score	0.996 0.997 0.996	0.987 0.995 0.990	0.990 0.979 0.983	0.981 0.965 0.971	0.924 0.953 0.931	0.917 0.957 0.960	0.919 0.948 0.924	0.979 0.987 0.983	0.968 0.973 0.967

Macro average used to calculate recall, precision, and F1-score of multiple classes; LW, level walking; US, upstairs walking; DS, downstairs walking.

**Table 8 sensors-23-00495-t008:** Confusion matrix of the detections of fall and physical activities based on data from both feet.

True Class	Predicted Class							
	Fall	Sit	Stand	Jump	Jog	LW	US	DS
Fall	479.03 457.38 479.19	0.03 2.88 0.28	0.44 4.78 0.09	0.06 13.28 0.13	0 0.06 0	0.31 1.19 0.13	0.09 0.09 0.03	0.03 0.34 0.16
Sit	0 5.16 0	385.16 410.84 434.00	94.84 64.00 46.00	0 0 0	0 0 0	0 0 0	0 0 0	0 0 0
Stand	0.19 6.16 0	70.63 25.75 17.63	404.00 443.03 460.00	0.38 3.09 2.38	0 0 0	4.81 1.97 0	0 0 0	0 0 0
Jump	0.50 42.50 0.25	0 0 0	0.03 16.78 0.44	462.50 420.66 475.84	9.59 0.09 3.13	5.72 0.06 0.25	1.06 0.06 0.03	0.59 0.09 0.06
Jog	0.41 1.25 0	0 0 0	0.16 0 0	17.84 3.78 6.09	446.47 453.78 458.66	2.69 2.69 1.69	7.84 2.38 12.31	4.28 16.13 1.25
Walk	0.22 6.09 0.06	0 0 0	5.69 1.28 2.19	3.78 1.19 0.72	4.59 3.75 7.09	461.34 442.75 463.66	3.19 19.97 4.56	0.88 4.97 1.72
US	0.50 16.19 0.25	0 0 0	0 3.84 1.56	10.50 0 2.34	3.78 0.13 10.59	8.47 67.00 19.78	455.06 350.88 444.25	1.69 41.97 1.22
DS	0 8.59 4.81	0 0 0	0 0 0	0.03 1.44 0.03	7.66 9.22 2.34	4.22 17.81 0.47	2.84 35.72 1.38	465.25 407.22 470.97

The number of each true class is 480 for each participant. The listed values are the average over 32 participants when inertial data and inertial parameters (1st row), center of pressure (2nd row) and foot pressures, center of pressure, inertial data, and parameters (3rd row) were, respectively, used. LW, level walking; US, upstairs walking; DS, downstairs walking.

**Table 9 sensors-23-00495-t009:** Computational complexity of the deep neural networks.

Input Features	Input Shape	Number of Weights	Computations (MFLOPs)
Foot pressures	22,300	462,238	82.9
Center of pressure (CoP)	4300	460,510	82.0
Foot pressures + CoP	26,300	462,622	83.1
Inertial data	12,300	461,278	82.4
Inertial parameters	6300	460,702	82.1
Inertial (data + parameters) =	18,300	461,854	82.7
Foot pressures + CoP + inertial (data + parameters)	44,300	464,350	83.9
CoP + inertial (data + parameters)	22,300	462,238	82.9

## Data Availability

The data presented in this study are available upon request from the corresponding author.

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
