# Peer review of "Deep Neural Network for the Detections of Fall and Physical Activities Using Foot Pressures and Inertial Sensing"

_sensors, 2023, doi:10.3390/s23010495_

Round 1

Reviewer 1 Report

My comments here are concerned solely with the organization of the manuscript. Consideration of these points will, I believe, lead to an improved report that better illustrates the key concepts and conclusions.

1.     It requires further explanation with regard to the Problem Definition, and more technical highlights, and kindly signify the broad objectives as a sub-section after the finish of the Introduction.

2.     Recent works are not given clearly in the paper. The authors can be summarised in a literature review in table format with their challenges and all.

3.     Write one section for the proposed work and algorithm.

4.     Compare the results obtained with the recent state of the art.

5.     Briefly can explain the features of the dataset and can provide the available link.

6.     What is the computational complexity of the proposed approach? Limitations also can be included in the conclusion section. 

7.     If possible, the authors should add other exciting works as there are many research papers in this similar research area. Like: 

-       A Review on Deep Learning Techniques for IoT Data

In a conclusion, the technical content is good. Therefore, the contribution of this article is also satisfactory. I am accepting an article with minor revision for publication in this journal.

Reviewer 2 Report

This manuscript proposed a novel approach for falling and physical activity detection based on foot pressure and inertia sensing, where a deep neural network was developed for the task of interest. To validate the performance of the proposed method, the experimental data from 32 participants with various physical activities were collected. The results demonstrated that the proposed deep network has the best performance for detecting fall, in terms of a number of evaluation indicators. Overall, the topic of this research is interesting, and the manuscript was well organised and written. The detailed commnets are summarised as follows.

1.       The contribution and innovation of the manuscript should be clarified clearly in abstract and introduction.

2.       Broaden and update literature review on deep learning or CNN in practical applications, such as image processing and data analysis. E.g. Vision-based concrete crack detection using a hybrid framework considering noise effect.

3.       The performance of deep neural network is heavily dependent on the setting of hyperparameters. How did the authors set the model parameters in this research to achieve the optimal prediction performance?

4.       A confusion matrix is suggested to be added for better demonstrating the classification performance of developed deep neural networks.

5.       Training time should also be considered as an evaluation metric, because it is very important for its practical implementation.

6.       How about the robustness of the proposed method against noise effect?

7.       The proposed deep neural network has not been presented convincingly about its advantages. A comparative study with other deep learning models is suggested.

8.       More future research should be included in conclusion part.

Reviewer 3 Report

The following comments must be carefully revised to improve the quality of the paper.

(1) The designing reasons of the neural network structure shown in Figure 2 should be described in detail, including the setting of the number of convolution cores, the use of residual blocks, etc.

(2) More technical details should be added to Section 2.3, such as the forwarded propagation and error back propagation process.

(3) The hyper-parameters setting before training and the whole training process are ignored.

(4) Some comparative experimental results should be supplemented in the experimental part to prove the effectiveness of the proposed method.

(5) Some signal samples should be displayed in the paper.

(6) The generalization and regularization of deep learning models across various scenes and conditions has not been discussed in the Introduction or Related Work.The following related work must be cited and analyzed in Section 1, including “Improvement of generalization ability of deep CNN via implicit regularization in two-stage training process,” IEEE Access, vol. 6, pp. 15844-15869, 2018. “Faster mean-shift: GPU-accelerated clustering for cosine embedding-based cell segmentation and tracking.” Medical Image Analysis 71 (2021): 102048. “Compound figure separation of biomedical images with side loss,” Deep Generative Models, and Data Augmentation, Labelling, and Imperfections. Springer, Cham, 2021. 173-183. “Pseudo RGB-D Face Recognition,” in IEEE Sensors Journal, vol. 22, no. 22, pp. 21780-21794, 15 Nov.15, 2022.

Reviewer 4 Report

Line 32 should include 2 simple references

The English grammar in the article can be significantly improved, it is recommended that the authors have the paper reviewed by a company that provides this service.

The introduction is well written and covers sufficient literature for the technical aspects in the field of research, however – in these proposed systems it is very important that human-centric design and usability are considered. While introducing related systems it is suggested that the authors include a brief description of the system usability for practical implementation in the real world.

Fall and PA descriptions on lines 105-123 can be better presented through a diagram or sectioned list.

Did the authors collect information on each participant’s shoe size? Is this accounted for in the FSR data collection and models?

Data list in lines 153-164 can be more clearly presented.

What is the sensing range of each FSR? Does this ever reach peak value during the experiments?

The selection of CNN is reasonable. A brief justification of why these particular methods were used compared to other methods would assist the reader.

Line 210 is confusing, please reword and check this sentence.

There is some confusion how the COPx and COPy are computed, is there a calibration frame taken into account and are there other references in literature that support this method of calculating COPx and COPy using a customized FSR array?

How was the FSR array constructed, what conductive and resistive materials are used? Are there any resistance variances between sensors that are distant from the connection points?

The Discussion section is very short and does not discuss the results from the experiments in enough detail. Please extend the Discussion section and provide more detail in connection with previous literature.

In addition, please include a section in the Discussion section that addresses the feasibility and usability of the proposed system in the real-world or how it could be adapted for real-world use.

Round 2

Reviewer 2 Report

All the technical issues have been well addressed by the authors, so I suggest this revised version can be accepted for publication.

Reviewer 3 Report

All the comments have been well revised and thus this paper can be accepted for publication.